# Prognostic Role of Lung Ultrasound in Children with Bronchiolitis: Multicentric Prospective Study

**DOI:** 10.3390/jcm11144233

**Published:** 2022-07-21

**Authors:** Laura Gori, Antonella Amendolea, Danilo Buonsenso, Stefano Salvadori, Maria Chiara Supino, Anna Maria Musolino, Paolo Adamoli, Alfina Domenica Coco, Gian Luca Trobia, Carlotta Biagi, Marco Lucherini, Alberto Leonardi, Giuseppe Limoli, Matteo Giampietri, Tiziana Virginia Sciacca, Rosa Morello, Francesco Tursi, Gino Soldati

**Affiliations:** 1Pediatric Unit, Valle del Serchio General Hospital, 55051 Barga, Italy; 2Pediatric Unit, Cecina Civil Hospital, 57023 Cecina, Italy; antonella.amendolea@uslnordovest.toscana.it; 3Department of Woman and Child Health and Public Health, Fondazione Policlinico Universitario A, Gemelli IRCCS, 00168 Rome, Italy; rosa.morello91@gmail.com; 4Institute of Clinical Physiology, CNR, 56124 Pisa, Italy; stefsa@ifc.cnr.it; 5Department of Pediatric Emergency, Bambin Gesù Children’s Hospital IRCCS, 00165 Rome, Italy; mariachiara.supino@opbg.net (M.C.S.); annamaria.musolino@opbg.net (A.M.M.); 6Pediatric Unit, Moriggia Pelascini Hospital, Gravedona et Uniti, 22015 Como, Italy; paolo.adamoli@gmail.com (P.A.); alfincoco@gmail.com (A.D.C.); 7Pediatric and Pediatric Emergency Room Unit, Cannizzaro Emergency Hospital, 95126 Catania, Italy; trobia@icloud.com (G.L.T.); tizianav.sciacca@gmail.com (T.V.S.); 8Pediatric Emergency Unit, Sant’Orsola Hospital IRCCS, 40138 Bologna, Italy; carlottabiagi@yahoo.it; 9Pediatric Unit, Nottola Hospital, Montepulciano, 53045 Siena, Italy; marco2.lucherini@uslsudest.toscana.it; 10Pediatric Clinic, Department of Surgical and Biomedical Sciences, University of Perugia, 06132 Perugia, Italy; alberto.leonardi88@gmail.com; 11Department of Pediatrics, Lodi Hospital, 26900 Lodi, Italy; giuseppe.limoli@asst-lodi.it; 12Department of Maternal and Child Health, Division of Neonatology and Neonatal Intensive Care Unit, S. Chiara Hospital, University of Pisa, 56100 Pisa, Italy; giampimg@gmail.com; 13Pneumology Unit, Civil Hospital, Codogno, 26845 Lodi, Italy; tursi_f@hotmail.com; 14Diagnostic and Interventional Ultrasound Unit, Valle del Serchio General Hospital, Castelnuovo Garfagnana, 55032 Lucca, Italy; soldatigino@yahoo.it; 15Pediatric Unit and Pediatric Emergency Unit, Azienda Ospedaliera Universitaria Policlinico San Marco, University of Catania, 95121 Catania, Italy; 16Pneumology Unit, Fondazione Policlinico Universitario A, Gemelli IRCCS, 00168 Rome, Italy

**Keywords:** bronchiolitis, lung ultrasound, children, respiratory syncytial virus, ultrasound score

## Abstract

There is increasing recognition of the role of lung ultrasound (LUS) to assess bronchiolitis severity in children. However, available studies are limited to small, single-center cohorts. We aimed to assess a qualitative and quantitative LUS protocol to evaluate the course of bronchiolitis at diagnosis and during follow-up. This is a prospective, multicenter study. Children with bronchiolitis were stratified according to clinical severity and underwent four LUS evaluations at set intervals. LUS was classified according to four models: (1) positive/negative; (2) main LUS pattern (normal/interstitial/consolidative/mixed) (3) LUS score; (4) LUS score with cutoff. Two hundred and thirty-three children were enrolled. The baseline LUS was significantly associated with bronchiolitis severity, using both the qualitative (positive/negative LUS *p* < 0.001; consolidated/normal LUS pattern or mixed/normal LUS *p* < 0.001) and quantitative models (cutoff score > 9 *p* < 0.001; LUS mean score *p* < 0.001). During follow-up, all LUS results according to all LUS models improved (*p* < 0.001). Better cut off value was declared at a value of >9 points. Conclusions: Our study supports the role of a comprehensive qualitative and quantitative LUS protocol for the identification of severe cases of bronchiolitis and provides data on the evolution of lung aeration during follow-up.

## 1. Introduction

Bronchiolitis is an acute lower respiratory tract infection affecting children under one [1] or two years of age [2], according to the Italian Consensus or to American Academy of Pediatrics (AAP) guidelines. It is the most common cause of non-elective hospitalization in these age groups [3,4,5,6].Diagnostic criteria for bronchiolitis are based on clinical history and examination [1,2,7]. Current guidelines do not suggest to routinely perform laboratory tests and, importantly, chest X-rays should be reserved for the most severe cases [2]. During the last decade, lung ultrasound (LUS) has been proved to be a useful diagnostic tool in many pediatric and neonatal diseases [8,9,10,11,12] and its application and utility in the management of bronchiolitis has also been reported [13,14,15,16,17,18,19,20,21,22,23,24,25,26,27,28,29,30]. 

However, available studies are limited by being mainly single-center studies that only assessed LUS scores during initial evaluation, using a limited bundle of LUS signs. Since bronchiolitis is a dynamic disease which may rapidly worsen in hours/days, a prospective longitudinal study addressing the role of LUS performed along multiple time-points is still highly needed. For these reasons, we performed a study aiming to define the prognostic role of LUS in children with bronchiolitis and its role in the short-term follow-up and monitoring of the disease. 

## 2. Patients and Methods

### 2.1. Population

This is an observational cohort and prospective, longitudinal and analytic study of a multicenter national cohort study, conducted between November 2018 and April 2020 in 12 Italian centers (Appendix A). 

We included children aged 0–12 months with a clinical diagnosis of bronchiolitis. After the first clinical examination, the severity of bronchiolitis was categorized by a severity score, as mild, moderate and severe illness according to Baraldi et al. [1], before the execution of the ultrasound exam. 

Exclusion criteria (based on clinical-anamnestic data, not on ultrasound findings), in analogy to the literature [13,14,15,16,17,18,19,20,21,22,23,24,25,26,27,28,29,30] were applied rigorously to avoid over fitting: children with immunosuppression, heart diseases, pneumonia (when the clinician according to clinical, laboratory and radiologic data classified the child has having pneumonia rather than bronchiolitis), neuromuscular diseases, cystic fibrosis, bronchopulmonary dysplasia (excluded according to the perinatal history, dependence of oxygen), positive history of foreign body inhalation, unstable critical conditions that required immediate life-saving procedures, and lack of parental consent. No control cases were enrolled because the aim of the study was to compare different stages of clinically diagnosed bronchiolitis. An individual data sheet for the collection of demographic, medical and clinical data, according to the clinical classification of bronchiolitis (Table 1), was used.

The study was approved by the Tuscan Meyer Committee (n 171/2018) and by the ethics committees of each participating center. For each patient enlisted, written consent was obtained from the parents.

### 2.2. Lung Ultrasound 

LUS was standardized both in terms of examination methods and timing. A 10 MHz (median frequency) linear probe, standard abdominal preset, mechanical index lower than 0.7, intermediate gain to obtain a pleural line defined but not too saturated, unique focus on the pleural line and depth 3–5 cm were used. The acquisitions were achieved by physicians having at least one year experience in pediatric LUS, using fundamental and harmonic images in relation to the technical characteristics of the ultrasound systems used by individual operators and in relation to the experience of the operator themself.

The first ultrasound was carried out within 3–6h of the first clinical assessment in each participating center. The second, third and fourth ultrasound were performed at 24, 72–96, and 144 h after first LUS.

Eight pulmonary fields were explored in each ultrasound (Figure 1). For each field, longitudinal and transverse scans were made (sixteen ultrasound scans total). Posterior lung fields were divided into paravertebral and basal as suggested in the literature by the studies of Basile and Taveira [14,17].

Each step of the clinical and ultrasound evaluation involved the dichotomous attribution (presence or absence) of ultrasound signs compatible with the diagnosis of bronchiolitis, the total grading of each area explored, and the number of involved areas. 

In accordance with literature data [31,32,33], which recognize vertical artifacts and consolidations in the bronchiolitis, the following classification of LUS findings was used: (Figure 2, Appendix A).
-A-lines: Horizontal reverbs of pleural line, normal ultrasound finding (Score 0 for the quantitative analysis, negative for the dycotomous analyses).-Short vertical artifacts (SVA, artifacts that do not reach the bottom of the screen, using a depth setting of 3 cm) According to a recent hypothesis [34], short artifacts may be produced by small channel or by acoustic traps which allows a quick release of the trapped acoustic energy [33]. Acoustic traps of this type are compatible with simple geometries such as those caused by minimal superficial alveolar collapses. Their microatelectatic nature and easy reversibility, in our opinion, justifies their distinction from the usual artifacts (B-Lines).-B-Lines showing a density of no more than 2 B lines per cm of the pleural line, with a depth setting of 3 cm, were described as Isolated B-Lines. Short vertical artifacts and Isolated B-Lines were considered clinically non-significant findings. Consequently, they were interpreted negative for dycotomous analyses, but received a score 1 for the quantitative analysis.-Multiple B lines (Lines B with a distance between them of less than half a cm to the confluence, remaining identifiable from each other) were considered pathological findings (positive), with a quantitative score of 2.-White lung (Subpleural field with various shades of white/gray without distinguishing lines B), pathological findings (positive), with a quantitative score of 3.

The choice of the ultrasound signs and the proposed scoring system was based on the following considerations. Although classifications of ultrasound signs of bronchiolitis in previous studies exist [13,14,15,16,17,22], there is no accepted consensus establishing their use for scoring. Consequently, we attributed value to what has been published in the clinical literature supported by recent technical studies [31,32,33,34,35,36] on the genesis and significance of the common ultrasound signs visible in bronchiolitis, especially in terms of lung tissue aeration, and on the possibility of these signs to semi-quantify the density of the peripheral lung. Further concepts will be explored in the discussion.

Based on these findings, four analysis models were created (Appendix A). Two qualitative models (dichotomous positive/negative relative to each area explored, and overall dichotomous positive/negative LUS examination), and a 4-level qualitative model based on the type of patterns (normal, interstitial, consolidative, mixed) was used; and two quantitative models (mean overall scores of the collected ultrasound findings and overall score > 9).

The ultrasound score of each patient was obtained from the sum of the individual scores of each involved area in that patient, considering the worst finding of each area. The logic of this increasing score was based on the physical and anatomical significance of the findings considered, in the hypothesis that each pattern represents an evolutionary degree of loss of lung ventilation from isolated B-Lines to consolidation, through the white lung [31,32,33].

Finally, the number of affected areas and the involvement of the paravertebral posterior fields were considered for the analysis of multivariate models, in consideration of their possible prognostic role [14].

LUS findings were reported on a model contained in the operator card, in order to reduce the operator variability as much as possible [37].

### 2.3. Statistical Analysis

Descriptive statistics were expressed as mean ± standard deviation or absolute frequency and percentage, where appropriate. For categorical variables, a univariate analysis was performed by χ^2^ test or the Fisher exact test, where appropriate, and comparisons between different time points were performed by McNemar test. For continuous variables, differences in means were tested by t-test for independent samples in the homoscedastic or the heteroscedastic version depending on the variabilities observed, and follow up comparisons were evaluated by paired samples t-test. Kolmogorov–Smirnov test was used to assess the normality of the data.

To identify the ecographic score cutoff value of 9, a receiver operating characteristic (ROC) curve analysis was performed and the area under the curve (AUC) and its 95% confidence interval (95% CI) was used to evaluate its performance. As the cutoff value was considered the one with the best performance, in term of both sensitivity and specificity, measured by the Youden index.

To quantify the effect of each ecographic pattern on predicting the diagnosis at t0, a logistic regression analysis was performed, and results were expressed by odds ratios (OR) and their 95% CI.

Both univariate and multivariate models were adopted. In univariate models, only the four different ecographic patterns were separately considered. In multivariate models, each ecographic pattern was adjusted by other ecographic patterns. In order to assess the predictive role of baseline LUS on the severity of bronchiolitis, four univariate logistic models and four multivariate logistic models were used. In univariate models, one of the four LUS patterns was considered as the only independent variable, while in multivariate models, in addition to the single LUS models, the number of fields involved, and the involvement of paravertebral fields were added as independent variables.

Multicollinearity was evaluated by the variance inflation factor (VIF) and considered relevant when VIF > 5.

For each model, predicted probabilities for each case were calculated and used in a ROC analysis. AUCs, as well as the sensitivities and specificities of the models and their 95% CI were also evaluated. AUCs were compared by the De Long test, and sensitivity and specificity 95% CIs were determined by the Jeffrey method.

Generalized Estimating Equation (GEE) models accounting for repeated measures were used to evaluate the effect of each ultrasonographic pattern on predicting the severity, considering all the four times observed. Moreover, within the repeated measure analysis, both univariate and multivariate models were adapted. 

Given that the number of cases of bronchiolitis treated in the specialized centers depends on the trend of respiratory virus epidemics, in designing the study the sample size was defined on the basis of the expected number of cases in the participating centers in the period of enrollment. In the pre-covid19 era, between 250 and 500 cases were expected to be observed.

Analysis was performed using SPSS Statistics for Windows, Version 23.0 (IBM SPSS, Statistics for Windows, Version 23.0. Armonk, NY, USA: IBM Corp.) and statistical significance was considered as *p* < 0.05.

## 3. Results

### 3.1. Study Population

A total of 233 patients (60.1% males, mean age 109.3 ± 86.1 days) were enrolled (Figure 3).The moderate/severe category were accurate in a single cohort due to the small number of severe forms. 160 (68.7%) had mild, 62 (26.6%) moderate, and 11 (4.7%) severe bronchiolitis. Demographic and clinical details are shown in Table 2. 

At the first ultrasound, patients with moderate/severe bronchiolitis had a worse ultrasound compared with those with mild bronchiolitis (Table 3). Children with moderate/severe bronchiolitis more frequently had a positive ultrasound, higher scores or scores higher than the cut-off value of 9, and more fields involved. Conversely, an early involvement of paravertebral areas was more frequent in mild cases.

### 3.2. Association between Initial LUS and Bronchiolitis Severity

LUS performed at baseline presentation was significantly associated with the severity of bronchiolitis when both qualitative patterns (positive vs. negative LUS; consolidated vs. normal LUS pattern or mixed vs. normal LUS) and quantitative models (using a cutoff score of 9 by increasing the value of the score) were used (Table 4). Better cut off value was declared a value > 9 points.

The multivariate models (Table 4), in which other LUS patterns have been considered, including the extension of lung disease (number of fields involved) and the early involvement of paravertebral fields, showed that only the quantitative score (OR 4.224; 95% CI 1.590–11.223; *p =* 0.004) and the extension of involved lung areas (OR 1.615; 95% CI 1.279–2.039; *p* < 0.0001) were significantly associated with bronchiolitis severity. Moreover, both in the positive/negative ultrasound qualitative models and in the 4-level ultrasound model, the statistically significant OR estimate was close to 1.6, indicating that for each extra lung field involved there was a 60% increase in the risk of moderate/severe bronchiolitis.

The ROC analyses show diagnostic capabilities of the different qualitative and quantitative to predict bronchiolitis severity (Figure 4, Table 5, Appendix A). 

Of the four AUC, although all statistically significant, the two related to qualitative patterns had low accuracy (0.6), having a high sensitivity of 0.918 for the positive/negative result and 0.877 for the 4-level LUS patterns, but low specificities, respectively 0.281 and 0.369. Conversely, the AUC related to the ultrasound score was higher and with values of 0.761 and 0.730, respectively, for the quantitative score and for the dichotomized score (cutoff 9), and showed a better balance between sensitivity, and specificity.

Adding to the ROC analysis the LUS variables “extension” and “early involvement of paravertebral areas” (Figure 3, Appendix A), the diagnostic ability of the models improved with the AUC that rose to 0.74, indicating greater diagnostic accuracy. For the two models, sensitivity reduction was observed to 0.658 for the positive/negative model and 0.644 for the 4-level model, with a significant increase in specificity rising to 0.725 and 0.731, respectively. 

Children with RSV bronchiolitis had statistically significant higher scores. Comparisons between bronchiolitis RSV+/RSV− children are shown in the Appendix A.

### 3.3. Role of LUS in the Monitoring of Bronchiolitis

One hundred and eighty-five (82.9%) were completely followed-up being enrolled at all scheduled controls. Changes of LUS qualitative and quantitative findings are detailed in Table 6 and Figure 5. Overall, LUS results gradually improved over time. In particular, considering the basal time and the last observation, the overall percentage of patients with moderate/severe bronchiolitis had gone from 26.5% to 4.3% (*p* < 0.001), the positive LUS passed from 80% to 53% (*p* < 0.001), the “mixed” result from 47.6% to 18.9% (*p* < 0.001). The percentage of patients with above-threshold scores decreased from 45.9% to 21.6% (*p* < 0.001) as had improved the average score and the average number of fields involved (9.5 to 6.2, *p* < 0.001; and 2.3 to 1.2, *p* < 0.001, respectively).

The univariate and multivariate logistic analyses comparing the different LUS modules at the last control with the severity of bronchiolitis during the same evaluation is shown in Appendix A. We found that the qualitative and quantitative scores still significantly correlated with the clinical picture, although the odds ratio was smaller compared to those at baseline (Table 3), supporting the evidence of an overall improvement of LUS findings.

## 4. Discussion

Our study found that qualitative and quantitative LUS findings aresuggestiveof an association with bronchiolitis severity and are useful in the follow-up of patients. To our knowledge, this is the largest prospective, multicenter study using a standardized follow-up and a well-established LUS protocol.

Although an association between LUS and bronchiolitis severity was already partially seen in previous studies [13,14,15,16,17,18,19,20,21,22,23,24,25,26], they were monocentric and included relatively small samples. More importantly, they included a combination of artifacts, not in line with recent advances in the field of LUS which highlight the potential of the multiple possible artifacts to be used to provide a semi-quantitative assessment of lung disease [32,33,35,36,38]. A quantitative approach can be more useful for personalized approaches to define disease severity and to quantify the changes of lung disease during follow-up. These aspects have been assessed in neonatal diseases [39], but never studied before in children with bronchiolitis. In our study, we used both quantitative and qualitative models that also included the analyses of different LUS patterns (interstitial, consolidative, mixed or normal), the early involvement of paravertebral areas and the extension of lung disease.

We found that children with moderate/severe bronchiolitis had a higher probability of having a positive LUS, higher scores and more lung areas involved. In addition, we found that the consolidative pattern indicated a three times greater probability of having moderate/severe bronchiolitis than those with a normal ultrasound pattern, and a mixed pattern of a 5-fold increase. The proposed gravity score was validated, as for every single point increase in the score there was an increase of 22.6% the risk of having moderate/severe bronchiolitis.

The ROC analyses allowed us to compare the accuracies of the qualitative and quantitative models. We found that the qualitative LUS models had a low diagnostic accuracy, being the good sensitivity [92%] of ultrasonography penalized by its low specificity. A better performance was achieved by introducing the ultrasound score and cutoff system (sensitivity and specificity of 77% and 69%, respectively). Interestingly, the accuracy of the model improved when we also included the number of thoracic fields involved (the extension of lung involvement), with a 60% increase in the risk of having a moderate/severe bronchiolitis for each further ultrasound field involved. Our findings suggest, therefore, that a comprehensive approach, that consider both the score and the extension, is necessary to provide a proper assessment of lung disease.

This comprehensive qualitative and quantitative approach needed to quantify severity of lung disease in children with bronchiolitis is in line with thepathophysiology of the disease. Childhood bronchiolitis determines structural alterations of the lung periphery due to altered ventilation related to the phenomenon of bronchiolar phlogosis, resulting in air trapping, dysventilation or atelectasis [40]. Since the ultrasound allows a bedside evaluation of the subpleural lung in terms of density, therefore of reduction [absolute or relative] of the quantitative ratio of the airspace and the interstice, it is reasonable to expect a combination of clinical and ultrasound parameters may be useful in clinical practice. In addition, positive associations between ultrasound and clinical aspects of bronchiolitis reinforce the underlying assumptions about the genesis of the artifact signs described in the literature, validating recent evidence [41,42,43,44]. In particular, there is increasing agreement that the B-lines are expression of focal areas (acoustic traps) [31] expressed by their density along the pleural line. Specifically, a worsening progression of the peripheral lung parenchyma seems to be characterized by the transitions from sporadic B lines, including short vertical artifacts, to multiple B lines, white lung and consolidations. White lung may be considered high-grade, preconsolidative lung density [35,36]. Therefore, in our opinion the distinction between white lung and small consolidation (generally surrounded by white lung) is not such as to vary their respective scores. These concepts have been confirmed in adult and are used to guide the alveolar recruitment maneuvers [39,44].

Although previous studies documented a prevalence of LUS findings in posterior paravertebral pulmonary fields [14], in our study the involvement of these areas was not associated with severity. Consolidation and white lung phenomena in the posterior areas can express only a greater tendency to collapse of these areas in subjects that are often placed in supine position, also in consideration of their rapid reversibility that we found during follow-up. In fact, this pattern has also been documented in other lung disease in young children [45].

Our study was the first one defining a standardized follow-up schedule providing information on LUS changes over time during bronchiolitis. The follow-up LUS assessments showed a progressive improvement of qualitative and quantitative LUS findings and a consensual reduction in the number of lung fields involved. This confirmed the general benignity of the disease and reinforces the hypothesis that much of the radiological alterations are supported by alterations of peripheral ventilation rather than to parenchymal phlogosis or alveolar consolidation. These findings can have clinical implications if confirmed on larger studies, since it can be speculated that the lack of rapid reversibility of LUS signs, especially consolidated, can suggest a bacterial overlap. The ultrasound follow-up also allows us to detect any complications of a bronchiolitis which usually has a benign self-limiting course, since if after an apparent clinical/instrumental improvement a worsening of the subject’s condition is observed, the ultrasound can show larger consolidations with tree-like bronchograms that can suggest possible bacterial complications, as suggested in other studies [19,46].

Another peculiarity of our study, compared with previously published ones, is the use of the quantitative score of SVA. As proved by recent literature, ultrasound is a good tool to define superficial pulmonary alterations. These occur with a flow of pattern ranging from isolated vertical artifacts (Lines B) to white lung, and mixed patterns with consolidation. Although SVA have not been widely described in the lung ultrasound literature, their genetic basis can be predictable, essentially representing acoustic traps capable of small vibrational responses [47,48]. We therefore consider it appropriate to consider them as phenomena of minor hypoventilation, easily reversible, and with less weight in the proposed score. In fact, in a previous study we found that healthy newborns, that have still immature lungs, have several SVA that reduce during the following months [49]. Since we documented them as healthy infants, we only counted the SVA for the quantitative score, but were considered as normal in the qualitative one. In subjects with bronchiolitis these phenomena are present in the majority of cases, they are multifocal and in great majority relative to alterations of the ventilation, considered their rapid reversibility.

Our study has limitations. First, the clinician performing LUS was not blinded to the clinical parameters. Second, we did not include parameters to assess and quantify the pleural line abnormalities. Third, the design of this study did not consider the comparison between echo and X-ray imaging, although more than half of the subjects with moderate/severe bronchiolitis underwent chest X-rays. Fourth, only one subject needed intensive care unit hospitalization. Fifth, the severity score used is not consensus based, but it is supported by recent theoretical hypothesis and experimental models [34,35,36,48,50]. Sixth, due to the emergency scenario related with COVID-19 pandemic and its impact on the circulation of respiratory viruses, the study ended earlier than planned in February 2020, possibly limiting the number of patients with severe bronchiolitis enrolled. Seventh, no control cases were enrolled because the aim of the study was to compare different stages of clinically diagnosed bronchiolitis. Last, a clarification is needed about some LUS features we have considered or described in the methods section. As there is not yet rigorous classification, standardization and interpretation of LUS features, we have tried to be more rigorous as possible by describing as better as possible each LUS feature (e.g., SVA or non-confluent B lines), with the final goal of allowing the same interpretation of each LUS sign by all the study members, which is particularly relevant in the context of a multicenter national study. Importantly, although some classifications may not be perfect, they are based on decades of clinical and physical studies of the experts of the Italian Academy of Thoracic Ultrasound, which trained in a standardized way all participants of this study. However, our study has several strengths: the number of patients enlisted is higher than the other studies published in the literature so far; it is a multicenter study; the inclusion of ultrasound follow-up until seven days after first enrollment; the introduction of a unique reporting template that allowed experimenters a more univocal interpretation of the LUS features; the ultrasound execution protocol was strictly defined a priori; the comparison between classes of the same disease which allowed to highlight the prognostic role of LUS.

## 5. Conclusions

In conclusion, our study found that a comprehensive LUS evaluation, which included a qualitative and quantitative assessment of LUS findings and the extension of the involved areas, does not have a strong association with moderate/severe bronchiolitis but it can still provide useful information to the clinician conducting the follow-up and to the management of bronchiolitis. Moreover, this approach allowed us to define the favorable evolution of lung disease in children with an uncomplicated course of bronchiolitis. These findings confirm the utility of LUS in the management of infants with bronchiolitis.

## Figures and Tables

**Figure 1 jcm-11-04233-f001:**
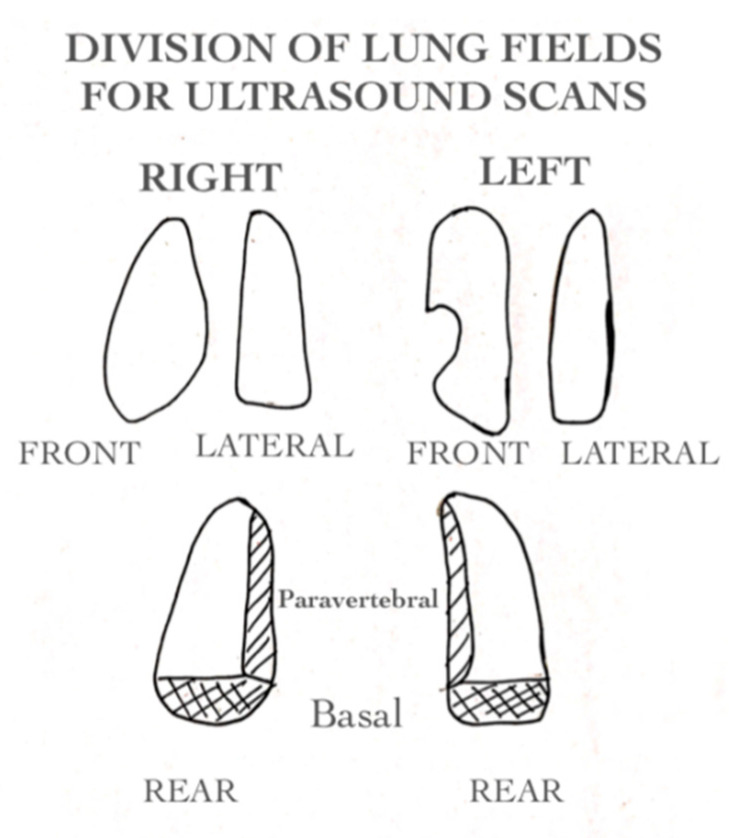
Division of lung fields for ultrasound scans. Eight pulmonary fields were explored in each ultrasound. For each field longitudinal and transverse scans were made for a total of sixteen ultrasound scans.

**Figure 2 jcm-11-04233-f002:**
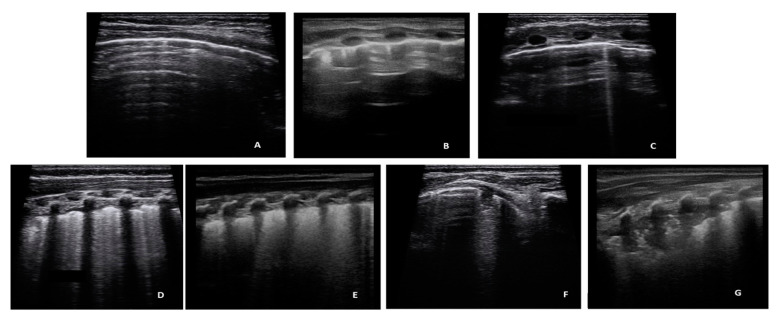
Images of the seven types of artifacts. (**A**) A-lines; (**B**) Short Vertical Artifacts; (**C**) Isolated B Lines; (**D**) Confluent B-Lines; (**E**) White Lung; (**F**) Sub-pleural consolidation ≤ 1 cm; (**G**) Sub-pleural consolidation > 1 cm.

**Figure 3 jcm-11-04233-f003:**
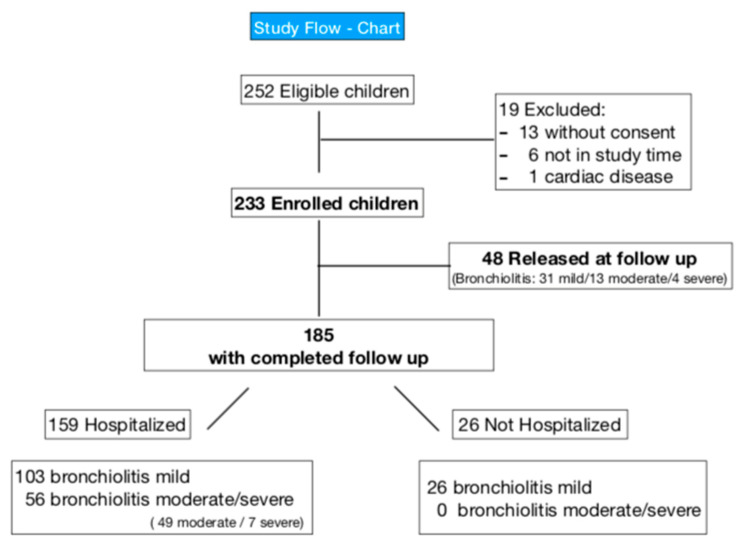
Study flow-chart.

**Figure 4 jcm-11-04233-f004:**
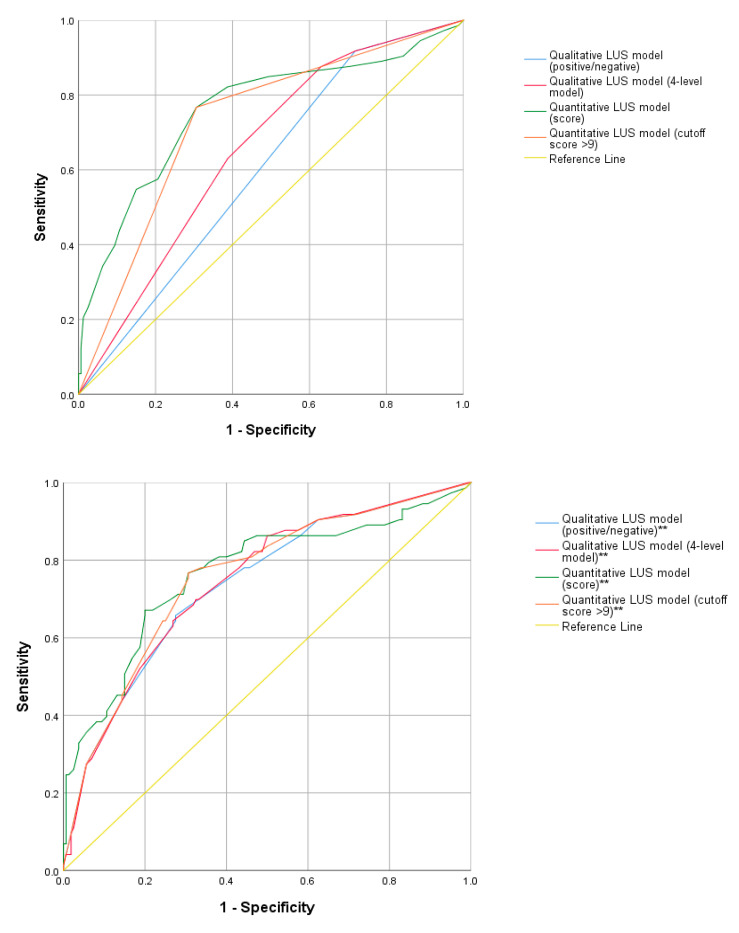
ROC curves comparing qualitative (positive/negative LUS, 4-level LUS including normal, interstitial, consolidative and mixed pattern) and quantitative (mean scores, cutoff score of 9) LUS models. ** ROC curves using the LUS models with the adjunction of the LUS variables “extension” and “early involvement of paravertebral areas”.

**Figure 5 jcm-11-04233-f005:**
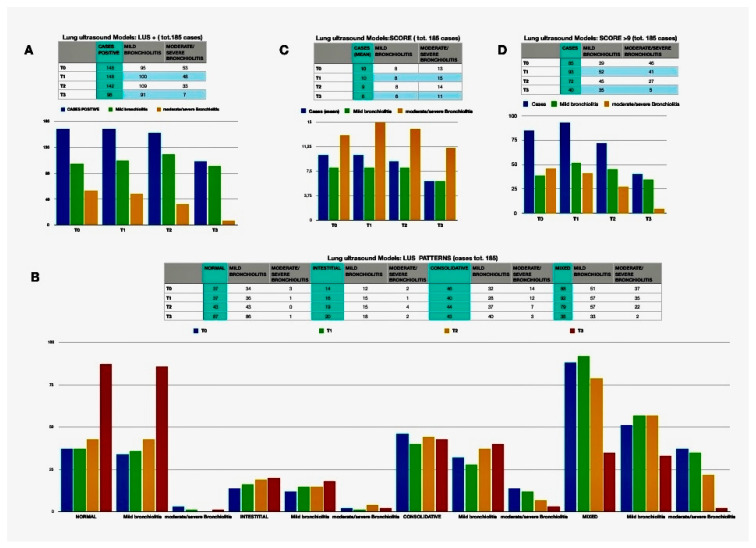
Lung Ultrasound (LUS) patterns at baseline and during follow-up. (**A**) shows changes in number of children with positive LUS; (**B**) shows changes in type of patterns; (**C**) shows changes in mean scores; (**D**) shows number of children with a LUS score higher than 9.

**Table 1 jcm-11-04233-t001:** Clinical criteria used to classify bronchiolitis according to Consensus Baraldi et al. (modified). The presence of two or more criteria from each category listed in the table attribute bronchiolitis to that severity class.

	MILD	MODERATE	SEVERE
**RESPIRATORY RATE**	Normal or slightly increased	Increased	Significantly increased.
**RESPIRATORY WORK**	Slight retractions of the chest wall	Modest retractions of the chest wall Sway of the head (nodding) Breath of the nasal fins	Significant retractions of the chest wall Grunting Breath of the nasal fins
**SATURATION OF OXYGEN**	O2 supplementation not required SatO2 > 95%	SatO2 90–95%	SatO2 < 90% No response to O2
**FEEDING**	Normal or slightly reduced	50–75% of normal food intake	<50% of normal food intake Inability to feed
**APNEA**	Absent	Brief episodes	Episodes on the rise

**Table 2 jcm-11-04233-t002:** Study Population. RSV: respiratory syncytial virus. * Home therapy before hospital admission (hypertonic solution and/or Bronchodilators and/or Cortisone for od).

	All (*n* = 233)	Bronchiolitis Mild (*n* = 160)	Bronchiolitis Moderate/Severe (*n* = 73)	*p* Value
**Characteristics**				
**Age**, days (SD)	109.3 (±86.1)	122.7 (±92.1)	80 (±62.3)	0.001
**Sex**				0.019
Female, *n*° (%)	93 (39.9%)	72 (45%)	21 (28.8%)	
Male, *n*° (%)	140 (60.1%)	88 (55%)	52 (71.2%)	
**Prematurity** (*n* = 230)				
Yes, *n*° (%)	20 (8.7%)	11 (7.0%)	9 (12.3%)	0.182
**History of difficulty in feeding** (*n* = 228)				
Yes, *n*° (%)	151 (66.2%)	92 (59.4%)	59 (80.8%)	0.001
* **Home therapy before hospital admission for three days** (*n* = 231)				
Yes, *n*° (%)	117 (50.6%)	85 (53.8%)	32 (43.8%)	0.159
**Rhinorrhea** (*n* = 231)				
Yes, *n*° (%)	227 (98.3%)	156 (98.7%)	71 (97.3%)	0.425
**Crackling noises on physical examination** (*n* = 230)				
Yes, *n*° (%)	198 (86.1)	134 (84.8%)	64 (88.9%)	0.407
**Wheezing** (*n* = 230)				
Yes, *n*° (%)	154 (67%)	107 (67.7%)	47 (65.3%)	0.715
**Fever** (*n* = 231)				
Yes, *n*° (%)	83 (35.9%)	52 (32.9%)	31 (42.5%)	0.159
**Intercostal retractions** (*n* = 231)				
Yes, *n*° (%)	164 (71%)	96 (60.8%)	68 (93.2%)	0.001
**Increased Respiratory Rate** (*n* = 230)				
Under 50 breath/min, *n*° (%)	75 (32.6%)	35 (22.3%)	40 (54.8%)	0.001
Between 51–60 breath/min, *n*° (%)	106 (46.1%)	95 (60.5%)	11 (15.1%)	
Above 61, breath/min	49 (21.3%)	27 (17.2%)	22 (30.1%)	
**Oxygen saturation** (*n* = 229)				
Under 92% in aa, *n*° (%)	39 (17%)	10 (6.4%)	29 (39.7%)	0.001
Between 93–95% in aa, *n*° (%)	45 (19.7%)	17 (10.9%)	28 (38.4%)	
Above 95% in aa, *n*° (%)	145 (63.3%)	129 (82.7%)	16 (21.9%)	
**RSV** (*n* = 216)				
Positive, *n*° (%)	126 (58.3%)	73 (51.0%)	53 (72.6%)	0.002
**Chest X-ray** (*n* = 228)				
No, *n*° (%)	160 (70.2%)			
Yes, *n*° (%)	68 (29.8%)	29 (18.7%)	39 (53.4%)	0.001
**Hospitalization**				
Yes, *n*° (%)	192 (82.4%)	119 (74.4%)	73 (100.0%)	0.001
**HFNC** (*n* = 225)				
Yes, *n*° (%)	50 (22.2%)	16 (10.5%)	34 (47.2%)	<0.001
**nCPAP** (*n* = 225)				
Yes, *n*° (%)	5 (2.2%)	0 (0%)	5 (6.9%)	0.001

**Table 3 jcm-11-04233-t003:** Results at the baseline lung ultrasound. Comparison of LUS findings between patients with moderate/severe and mild bronchiolitis.

LUNG ULTRASOUND MODELS		All Cases (*n* = 233)	Mild Bronchiolitis (*n* = 160)	Moderate/ Severe Bronchiolitis (*n* = 73)	*p* Values
**Ultrasound positive vs. negative**	**Ultrasound positive, *n*° (%)**	182 (78.1)	115 (71.9)	67 (91.8)	0.001
	**Ultrasound negative, *n*° (%)**	51 (21.9)	45 (28.1)	6 (8.2)	
**Qualitative result ultrasound**	**Normal, *n*° (%)**	51 (21.9)	45 (28.1)	6 (8.2)	0.001
	**Interstitial, *n*° (%)**	17 (7.3)	14 (8.8)	3 (4.1)	
	**Consolidative, *n*° (%)**	57 (24.5)	39 (24.4)	18 (24.7)	
	**Mixed *n*° (%)**	108 (46.4)	62 (38.8)	46 (63)	
**Score**, mean (SD)		9.4 (5.2)	7.9 (4.3)	12.8 (5.6)	0.001
**Score** **cut off > 9,** *n* (%)		105 (45.1%)	49 (30.6%)	56 (76.7%)	0.001
**Lung fields number, mean (SD)**		2.2 (1.8)	1.7 (1.6)	3.3 (1.9)	0.001
**Early involvement of the paravertebral field on LUS**	No, *n*° (%)	80 (34.3%)	69 (43.1%)	11 (15.1%)	<0.001
	Yes, *n*° (%)	153 (65.7%)	91 (56.9%)	62 (84.9%)	

**Table 4 jcm-11-04233-t004:** Multivariate models showing correlation of the different lung ultrasound models at baseline with bronchiolitis severity.

Model	Variables in the Model		OR	95% CI		*p* Value
**Univariate models**	**Positive ultrasound**	Yes	**4.370**	1.770	10.785	0.001
		No	1			
	**Ultrasound result**	Interstitial	1.607	0.355	7.276	0.538
		Consolidative	**3.462**	1.250	9.586	**0.017**
		Mixed	**5.565**	2.188	14.150	<0.001
		Normal	1			
	**Score**		**1.226**	1.146	1.311	<0.001
	**Score > 9**	Yes	**7.462**	3.941	14.129	<0.001
		No	1			
**MULTIVARIATE ANALYSES USING DIFFERENT LUS MODELS** **Multivariate models**	**MULTIVARIATE MODELS BASED ON QUALITATIVE LUS (positive vs. negative) without or with other LUS variables (Early involvement of paravertebral lung fields and extension)**					
	**Positive ultrasound**	Yes	0.699	0.180	2.721	0.605
		No	1			
	**Early involvement of paravertebral lung fields**	Yes	1.622	0.548	4.801	0.383
		No	1			
	**Number of fields involved**		**1.615**	1.279	2.039	**0.000**
	**MULTIVARIATE MODELS BASED ON QUALITATIVE LUS (type of lung disease) without or with other LUS variables (Early involvement of paravertebral lung fields and extension)**					
	**Ultrasound result**	Interstizial	0.470	0.078	2.846	0.411
		Consolidative	0.770	0.193	3.066	0.710
		Mixed	0.691	0.156	3.067	0.627
		Normal	1			
	**Early involvement of lung fields**	Yes	1.711	0.569	5.144	0.339
		No	1			
	**Number of fields involved**		**1.594**	1.239	2.050	<0.001
	**MULTIVARIATE MODELS BASED ON QUANTITATIVE LUS (mean score) without or with other LUS variables (Early involvement of paravertebral lung fields and extension)**					
	**Score**		**1.213**	1.120	1.313	<0.001
	**Early involvement of lung fields**	Yes	1.241	0.515	2.992	0.630
		No	1			
	**MULTIVARIATE MODELS BASED ON QUANTITATIVE LUS (score > 9) without or with other LUS variables (Early involvement of paravertebral lung fields and extension)**					
	**Score > 9**	Yes	**4.224**	1.590	11.223	**0.004**
		No	1			
	**Early involvement of lung fields**	Yes	1.043	0.397	2.739	0.932
		No	1			
	**Number of fields involved**		1.220	0.928	1.604	0.155

**Table 5 jcm-11-04233-t005:** ROC Models. Univariate and multivariate models.

Model		AUC	*p* Value	95% CI		Sensitivity	95% CI		Specificity	95% CI		LR +	95% CI		LR −	95% CI	
**Univariate models**	**Positive ultrasound**	0.600	0.015	0.52	0.67	0.918	0.838	0.965	0.281	0.216	0.354	1.3	1.1	1.4	0.3	0.1	0.7
	**Ultrasound 4 level**	0.654	<0.001	0.58	0.72	0.877	0.787	0.937	0.369	0.297	0.445	1.4	1.2	1.6	0.3	0.2	0.6
	**Score**	0.761	<0.001	0.68	0.83	0.767	0.661	0.852	0.694	0.619	0.761	2.5	1.9	3.3	0.3	0.2	0.5
	**Score > 9**	0.730	<0.001	0.66	0.80	0.767	0.661	0.852	0.694	0.619	0.761	2.5	1.9	3.3	0.3	0.2	0.5
**Multivariate models**	**Positive Ultrasound**	0.738	<0.001	0.66	0.80	0.658	0.544	0.759	0.725	0.652	0.790	2.4	1.8	3.2	0.5	0.3	0.7
	**Ultrasound 4 level**	0.743	<0.001	0.67	0.81	0.644	0.530	0.746	0.731	0.659	0.795	2.4	1.8	3.3	0.5	0.4	0.7
	**Score**	0.762	<0.001	0.69	0.83	0.767	0.661	0.852	0.700	0.626	0.767	2.6	2.0	3.3	0.3	0.2	0.5
	**Score > 9**	0.755	<0.001	0.68	0.82	0.767	0.661	0.852	0.694	0.619	0.761	2.5	1.9	3.3	0.3	0.2	0.5

**Table 6 jcm-11-04233-t006:** Changes in lung ultrasound (LUS) models during the different subsequent controls (T0, T1, T2 and T3), and changes in treatments offered. HFNC: high flow nasal cannulae. nCPAP: nasal Continuous Positive Airway Pressure.

			TO			T1			T2			T3			*p* Value
			CASES (*n* = 185)	Mild Bronchilitis (*n* = 129),69.7%	Moderate/Severe Bronchiolitis (*n* = 56), 30.3%	CASI (*n* = 185)	Mild bronchilitis (*n* = 136), 73.5%	Moderate/severe Bronchiolitis (*n* = 49), 26.5%	CASES (*n* = 185)	Mild Bronchiolitis (*n* = 152), 82.2%	Moderate/Severe Bronchiolitis (*n* = 33), 17.8%	CASES (*n* = 185)	Mild bronchilitis (*n* = 177), 95.7%	Moderate/severe Bronchiolitis (*n* = 8), 4.3%	
**LUNG ULTRASOUND MODELS** **LUS MODELS**	**LUS +**		148 (80%)	95 (73.6%)	53 (94.6%)	148 (80%)	100 (73.5%)	48 (98%)	142 (76.8%)	109 (71.7%)	33 (100%)	98 (53%)	91 (51.4%)	7 (87.5%)	<0.001
	**LUS PATTERNS**	**NORMAL**	37 (20%)	34 (26.4%)	3 (5.4%)	37 (20%)	36 (26.5%)	1 (2.0%)	43 (23.2%)	43 (28.3%)	0	87 (47%)	86 (48.6%)	1 (12.5%)	<0.001
		**INTESTITIAL**	14 (7.6%)	12 (9.3%)	2 (3.6%)	16 (8.6%)	15 (11%)	1 (2%)	19 (10.3%)	15 (9.9%)	4 (12.1%)	20 (10.8%)	18 (10.2 %)	2 (25%)	
		**CONSOLIDATIVE**	46 (24.9%)	32 (24.8%)	14 (25.0%)	40 (21.6%)	28 (20.6%)	12 (24.5%)	44 (23.8%)	37 (24.3%)	7 (21.2%)	43 (23.2%)	40 (22.6%)	3 (37.5%)	
		**MIXED**	88 (47.6%)	51 (39.5%)	37 (66.1%)	92 (49.7%)	57 (41.9%)	35 (71.4%)	79 (0.427%)	57 (0.375%)	22 (0.667%)	35 (0.189%)	33 (0.186)	2 (0.25%)	
	**SCORE,** mean ± SD		9.5 ± 5.2	8.0 ± 4.4	13.2 ± 5.0	10.0 ± 5.9	8.3 ± 5.1	14.8 ± 5.7	8.8 ± 5.1	7.7 ± 4.6	13.6 ± 4.5	6.2 ± 4.5	6.0 ± 4.4	10.5 ± 5.8	<0.001
	**Score > 9**		85 (45.9%)	39 (30.2%)	46 (82.1%)	93 (50.3%)	52 (38.2%)	41 (83.7%)	72 (38.9%)	45 (29.6%)	27 (81.8%)	40 (21.6%)	35 (19.8%)	5 (62.5%)	<0.001
	**Extension** ± SD		2.3 ± 1.8	1.8 ± 1.6	3.4 ± 1.8	2.5 ± 2.0	1.9 ± 1.8	4.1 ± 1.8	2.1 ± 1.8	1.8 ± 1.6	3.6 ± 1.6	1.2 ± 1.5	1.1 ± 1.4	2.9 ± 2.0	<0.001

## Data Availability

Available upon request.

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
