# Peer review of "Prognostic Role of Lung Ultrasound in Children with Bronchiolitis: Multicentric Prospective Study"

_jcm, 2022, doi:10.3390/jcm11144233_

Round 1

Reviewer 1 Report

This is an important article, in this reviewer’s opinion. It tries to answer a very relevant question regarding the accuracy of lung ultrasound (LUS) in the estimation of bronchiolitis prognosis. And the importance relies, contrary of what the authors write since two times in the entire text (and sounds a bit presumptuous…), not in being the “first study” but in being the study with the biggest sample size trying to answer rigorously this scientific question. Because it represents a huge scientific effort due to its multicentric nature. Studying the role of echography in stablishing the prognosis of bronchiolitis is, in my opinion an appropriate, very pertinent study that remains yet to be done. So I think it must be published in JCM, but only after some corrections have been done to improve the understanding of the real results of authors´ analysis. A priori these results might influence the normal clinical practice all over the world.

The introduction is well written. Concisely review the state of art, and in the last sentence precisely states the main objective as the final result of a logic argument. I would change “using a limited cohort of LUS signs” for “using a limited bundle of LUS signs” or something like that: cohort has a very precise technical meaning in the medical literature. And I would also drop the words …”a multicentric, prospective, longitudinal observational”… that are technical words about the experimental design of the investigation, and leave only the phrase this way: …”, we performed a study aiming to define the prognostic role of LUS in children with bronchiolitis and its role in 67 the short-term follow-up and monitoring of the disease”. This is the actual objective of the study, written in substantive form not using wording of the methods section.

The text of the ”Materials and methods” section sounds well for me. In it, authors describe the population (inclusion/exclusion criteria), the sampling, the ethical details and the experimental design. I think here (line 71) fit better the words: “prospective, longitudinal and analytic study of a multicentric national cohort study, conducted...”. The only detail to fix is to change the title of the section “Materials and Methods” for “Patients and Methods”, more often used in the medical papers, because the sampling is done with people.

From my point of view, is “Results” section that needs some re-building. Let me try to suggest some changes, in order to gain clarity and ease the reader understanding.

First: in my opinion, the most important result of all analysis is buried in the supplementary material!. It is Supplementary Table S4, recording the areas under AUCs (discriminative capacity) and Sensitivity / Specificity (diagnostic accuracy for a worst prognosis) with their 95% Confidence intervals of all the adjusted models. For me, it must be the actual Table 4 of the main text, and it would be important to record also the likelihood ratios (with their 95%CI) of every model. Please, incorporate this information to the main text or the paper. Is this information that reader needs to incorporate in daily clinical scenario.

Instead of this, authors might place in the Supplementary material the “Table 4. Comparison between bronchiolitis VRS +/-.” Data in here is only an explanation confirming information stated in the sentence of line 256-257 of page 11. It may be at the supplement. It might be the new Table S4.

Second: the study flow-chart and the table 1 are correct. Probably the phrase( hypertonic solution and/or Bronchodilators and/or Cortisone for od )” might better be located in the legend of the table, after the RSV, as the explanation of an asterisk located inside the table.

Third: Table 2, with the results of first ultrasound is correct. In the text of page 7 (line 205 to 210) I would drop the numbers and the p-values because are redundant with the information in table 2 and make difficult the reading.

Four: Association between LUS and bronchiolitis severity. I would name this part “Association between initial LUS and bronchiolitis severity”, in order to clarify. An again, the numbers and p-values of text of lines 215 – 233 must be dropped because are redundant with the information in table 3. And please, underscore that better cut off value was declared a value > 9 points. This is an important issue for the posterior application in the clinical arena of the reader.

Five: Figure 3 is very confusing in this actual draft. I think it would be better idea to split tha graph in two graphs, one with the ROC curves without the addition of LUS variables and other with the addition of LUS variables (those ROCs now signaled with the two asterisk). And in the text of the beginning of page 11, authors might drop again the numbers and p-values that are redundant of the information of Table 4 (that in the new draft must be the Table S4, completed with the LRs), maintaining only the sustantive wording. One important detail that actual text (and table S4) lacks is the statistical significance of THE DIFFERENCES between the different ROCs curves. For instance, in lines 250 and 251 the texts says: “Adding to the ROC analysis the LUS variables “extension” and “early involvement of paraverte-bral areas”, the diagnostic ability of  of the models improved with the AUC that rose to 0.74, indicating greater diagnostic accuracy”. But this must be confirmed computing the p-value between the difference of this AUC (0.74) with the former AUC (without the LUS variables). Neither in Figure 3 nor in the actual Table S4 is computed the p-value of the difference between ROCs. It is computed the p-value against the null hypothesis of an area of 0.5. Authors must be precise and rigorous in this important affirmation.

In line 249-255 numbers and p-values must be removed because redundancy with the tables.

Six: Role of LUS in the monitoring of bronchiolitis. Table 5 is also correct, but plenty of numbers… Possibly will look better if changed to horizontal landscape frame. Please, remove again numbers and p-values that are redundant with table 5. And when it comes to Figure 4, I would propose that authors change the aesthetics for the codification of the categoric variables. In my opinion, reader will understand better to use the Y axis as a TIME-FRAMING axes (T0, T1, T2, T3, T4), and use the color code to describe the LUS patterns. For me is easy to compare the information gathered in clusters determined by temporal moments.

Seven: I think Table 6 might be better accommodated in the Supplement. The text relative to this table must remain in the main paper.

Finaly, in the “Discusion” section authors declare the accuracy of LUS findings in establishing the prognosis of bronchiolitis, and in the “Conclusions” they state that LUS evaluation has a strong association with bronchiolitis severity. Both assertions are NOT supported by their data in this investigation and must be softened. In the actual table S4, that must bel the new table 4, univariate models have mean Sensitivities that range from 0.92 to 0.77 with mean Specificities that range from 0.28 to 0.69, and multivariate models have mean Sensitivities that range from 0.66 to 0.77 with mean Specificities that range from 0.73 to 0.79. This numbers yield mean LR+ in univariate models that range from 1.3 to 2.5 (Weight of Evidence + = 1 to 4 decibans) and mean LR- that range from 0.2 to 0.3 (Weight of Evidence - = -5.3 to -4.8 decibans), and in multivariate models that range from 2.4 to 2.6 (Weight of Evidence + = 3.8 to 4 decibans) and mean LR- that range from 0.3 to 0.5 (Weight of Evidence - = -3 to -5 decibans). This is very far from been “strong association” in the accuracy of establishing the prognosis of bronchiolitis.

With all this changes done, the paper would be published in the journal.

Thank you for given me the opportunity of reviewing this important investigation.

Author Response

Dear Editor and reviewers

Thank you very much for your detailed revisions aimed at improving our paper.

Please find below a point-by-point response to your observations, changes have been highlighted in the main manuscript

REVIEWER 1

This is an important article, in this reviewer’s opinion. It tries to answer a very relevant question regarding the accuracy of lung ultrasound (LUS) in the estimation of bronchiolitis prognosis. And the importance relies, contrary of what the authors write since two times in the entire text (and sounds a bit presumptuous…), not in being the “first study” but in being the study with the biggest sample size trying to answer rigorously this scientific question. Because it represents a huge scientific effort due to its multicentric nature. Studying the role of echography in stablishing the prognosis of bronchiolitis is, in my opinion an appropriate, very pertinent study that remains yet to be done. So I think it must be published in JCM, but only after some corrections have been done to improve the understanding of the real results of authors´ analysis. A priori these results might influence the normal clinical practice all over the world.

The introduction is well written. Concisely review the state of art, and in the last sentence precisely states the main objective as the final result of a logic argument. I would change “using a limited cohort of LUS signs” for “using a limited bundle of LUS signs” or something like that: cohort has a very precise technical meaning in the medical literature. And I would also drop the words …”a multicentric, prospective, longitudinal observational”… that are technical words about the experimental design of the investigation, and leave only the phrase this way: …”, we performed a study aiming to define the prognostic role of LUS in children with bronchiolitis and its role in 67 the short-term follow-up and monitoring of the disease”. This is the actual objective of the study, written in substantive form not using wording of the methods section.

The text of the ”Materials and methods” section sounds well for me. In it, authors describe the population (inclusion/exclusion criteria), the sampling, the ethical details and the experimental design. I think here (line 71) fit better the words: “prospective, longitudinal and analytic study of a multicentric national cohort study, conducted...”. The only detail to fix is to change the title of the section “Materials and Methods” for “Patients and Methods”, more often used in the medical papers, because the sampling is done with people.

Thanks for the suggestion, we corrected.

From my point of view, is “Results” section that needs some re-building. Let me try to suggest some changes, in order to gain clarity and ease the reader understanding.

First: in my opinion, the most important result of all analysis is buried in the supplementary material!. It is Supplementary Table S4, recording the areas under AUCs (discriminative capacity) and Sensitivity / Specificity (diagnostic accuracy for a worst prognosis) with their 95% Confidence intervals of all the adjusted models. For me, it must be the actual Table 4 of the main text, and it would be important to record also the likelihood ratios (with their 95%CI) of every model. Please, incorporate this information to the main text or the paper. Is this information that reader needs to incorporate in daily clinical scenario.

Thanks for the suggestion, we have moved the S4 table in the paper  (now  table 5) and made the requested information additions.

Instead of this, authors might place in the Supplementary material the “Table 4. Comparison between bronchiolitis VRS +/-.” Data in here is only an explanation confirming information stated in the sentence of line 256-257 of page 11. It may be at the supplement. It might be the new Table S4.

Thanks for the suggestion, we have moved the table to supplementary materials ( now  table S4).

Second: the study flow-chart and the table 1 are correct. Probably the phrase “( hypertonic solution and/or Bronchodilators and/or Cortisone for od )” might better be located in the legend of the table, after the RSV, as the explanation of an asterisk located inside the table.

Thanks for the suggestion, we corrected ( now table 2).

Third: Table 2, with the results of first ultrasound is correct. In the text of page 7 (line 205 to 210) I would drop the numbers and the p-values because are redundant with the information in table 2 and make difficult the reading.

Thanks for the suggestion, we corrected

Four: Association between LUS and bronchiolitis severity. I would name this part “Association between initial LUS and bronchiolitis severity”, in order to clarify. An again, the numbers and p-values of text of lines 215 – 233 must be dropped because are redundant with the information in table 3. And please, underscore that better cut off value was declared a value > 9 points. This is an important issue for the posterior application in the clinical arena of the reader.

Thanks for the suggestion, we corrected

Five: Figure 3 is very confusing in this actual draft. I think it would be better idea to split tha graph in two graphs, one with the ROC curves without the addition of LUS variables and other with the addition of LUS variables (those ROCs now signaled with the two asterisk). And in the text of the beginning of page 11, authors might drop again the numbers and p-values that are redundant of the information of Table 4 (that in the new draft must be the Table S4, completed with the LRs), maintaining only the sustantive wording. One important detail that actual text (and table S4) lacks is the statistical significance of THE DIFFERENCES between the different ROCs curves. For instance, in lines 250 and 251 the texts says: “Adding to the ROC analysis the LUS variables “extension” and “early involvement of paraverte-bral areas”, the diagnostic ability of  of the models improved with the AUC that rose to 0.74, indicating greater diagnostic accuracy”. But this must be confirmed computing the p-value between the difference of this AUC (0.74) with the former AUC (without the LUS variables). Neither in Figure 3 nor in the actual Table S4 is computed the p-value of the difference between ROCs. It is computed the p-value against the null hypothesis of an area of 0.5. Authors must be precise and rigorous in this important affirmation.

Thanks for the suggestion, we have moved the S4 table in the paper  ( table 4) and made the requested information additions and we have added supplementary materials t Supplementary Table S4: The comparison between the AUC of the 2 qualitative models without and with the other LUS evidences

In line 249-255 numbers and p-values must be removed because redundancy with the tables.

Thanks for the suggestion, we corrected

Six: Role of LUS in the monitoring of bronchiolitis. Table 5 is also correct, but plenty of numbers… Possibly will look better if changed to horizontal landscape frame. Please, remove again numbers and p-values that are redundant with table 5. And when it comes to Figure 4, I would propose that authors change the aesthetics for the codification of the categoric variables. In my opinion, reader will understand better to use the Y axis as a TIME-FRAMING axes (T0, T1, T2, T3, T4), and use the color code to describe the LUS patterns. For me is easy to compare the information gathered in clusters determined by temporal moments.

Thanks for the suggestion, we corrected figure 4

Seven: I think Table 6 might be better accommodated in the Supplement. The text relative to this table must remain in the main paper.

Thanks for the suggestion, we have moved the table 6 in Supplementary ( table S6).

Finaly, in the “Discusion” section authors declare the accuracy of LUS findings in establishing the prognosis of bronchiolitis, and in the “Conclusions” they state that LUS evaluation has a strong association with bronchiolitis severity. Both assertions are NOT supported by their data in this investigation and must be softened. In the actual table S4, that must bel the new table 4, univariate models have mean Sensitivities that range from 0.92 to 0.77 with mean Specificities that range from 0.28 to 0.69, and multivariate models have mean Sensitivities that range from 0.66 to 0.77 with mean Specificities that range from 0.73 to 0.79. This numbers yield mean LR+ in univariate models that range from 1.3 to 2.5 (Weight of Evidence + = 1 to 4 decibans) and mean LR- that range from 0.2 to 0.3 (Weight of Evidence - = -5.3 to -4.8 decibans), and in multivariate models that range from 2.4 to 2.6 (Weight of Evidence + = 3.8 to 4 decibans) and mean LR- that range from 0.3 to 0.5 (Weight of Evidence - = -3 to -5 decibans). This is very far from been “strong association” in the accuracy of establishing the prognosis of bronchiolitis.

Thanks for the suggestion, we have moved the text.

With all this changes done, the paper would be published in the journal.

Thank you for given me the opportunity of reviewing this important investigation.

Thanks a lot for your support

Reviewer 2 Report

A brief summary

This observational, prospective, longitudinal, multicenter study aims to evaluate the role of lung ultrasound (LUS) in the assessment of bronchiolitis severity in children under 12 months of age. It shows the association between sonographic and clinical assessment of bronchiolitis severity short after clinical diagnosis (baseline) and at three subsequent time points during follow-up using four different LUS protocols/models (2 qualitative and 2 quantitative). It also shows the role of LUS in monitoring bronchiolitis and the diagnostic capabilities of these 4 LUS models to predict bronchiolitis severity. 

This is the first publication concerning LUS in bronchiolitis, including such a large number of patients (233 at the baseline and 185 at follow-up). To my knowledge, this is the longer follow-up time among studies published so far. LUS protocols were standardised between 12 pediatric centres. 

General concepts comments (Major criticism)

I have no major criticism.

Specific comments (Minor criticisms)

Abstract

"for the identification of severe cases of bronchiolitis". The presented study does not identify severe cases of bronchiolitis, as it assesses pulled moderate/severe cases compared to mild cases. Please rephrase it so that it represents the study's actual results. 

- "During follow-up, all LUS models improved" -> "During follow-up, LUS results according to all LUS models/protocols improved"

- It is essential to mention in the abstract the prognostic value of LUS, which was assessed in the study. 

Introduction

- "limited cohort of LUS signs"- What do the Authors mean by that? 

- "LUS has been proposed" - > "has been proved to be"

- There is one more recent publication on LUS in bronchiolitis, which should be mentioned: La Regina DP, Bloise S, Pepino D, Iovine E, Laudisa M, Cristiani L, Nicolai A, Nenna R, Mancino E, Di Mattia G, Petrarca L, Matera L, Frassanito A, Midulla F. Lung ultrasound in bronchiolitis. Pediatr Pulmonol. 2021 Jan;56(1):234-239. doi: 10.1002/ppul.25156. Epub 2020 Nov 24. PMID: 33151023.

Materials and methods

- The assessment of bronchiolitis severity is not precisely described in the main text. According to Supplementary Table 2, the assessment was performed according to the modified Italian consensus method (by Baraldi et al.) and not the Baraldi method itself. Rephrase, please. I also suggest that Supplementary table 2 should be incorporated into the main text as it is crucial for understanding the methods. 

- It is essential to describe that moderate and severe bronchiolitis are pulled together and why. 

- line 124-125: What about consolidations of exact 1 cm size? To be precise, a division should be used: consolidation ≤ 1 cm and consolidation > 1 cm.

- First Figure in the main text does not have either a title or a legend. Please, complete these elements.

- "No control cases…" – It belongs to the discussion. 

- "Standard abdominal preset" usually contains harmonic and compound imaging, while for lung ultrasound, they should be switched off to allow assessment of artefacts. 

- What does "the first contact with the recruiting centre" mean? It is not described what a "recruiting centre" is. 

- The lung fields are not clearly defined in the article's main text. Neither they are marked precisely in Supplementary Figure 1. The figure is not easy to interpret and understand. I suppose the lung fields are frontal right, frontal left, lateral right, lateral left, posterior paravertebral right, posterior paravertebral left, posterior basal right, and posterior basal left. However, it should be clear for the reader. The green and red fields representing longitudinal and transverse scans suggest that they were done on different parts of the lungs, while from the text, it is clear that transverse and longitudinal scans were both performed on each of the 8 lung fields. I suggest omitting these symbols. It is not described why paravertebral and basal areas are marked differently in the figure. I suggest omitting this check mark or explaining it. In my opinion, Supplementary Figure 1 should be incorporated into the main text, as it is crucial to understand the methods.

- "Each step of the clinical and ultrasound evaluation involved the dichotomous attribution (presence or absence) of ultrasound signs compatible with the diagnosis of bronchiolitis, the total grading of each area explored, and the number of involved areas." – It is not clear to the reader. Rephrase the sentence, please. What does it mean that the signs were compatible with the diagnosis of bronchiolitis? All the examined patients were diagnosed with bronchiolitis. 

- Short vertical artefacts were mentioned in neither of the cited publications (30-32). I do not recognise the 3 cm depth criterion. What is its source? Short vertical artefacts are mentioned in the ref. 48 (Demi 2022).

- Intercostal space is not always equal to 1 cm, especially in small children, including infants. If you use such a criterion, you should justify it. 

- I do not recognise the criterion "less than half cm" for multiple B-lines. What is the source of this criterion?

- The paragraph between lines 129 and 136 belongs to the discussion.

- line 130: "classifications of ultrasound signs of bronchiolitis" – reference 9. does not consider bronchiolitis (it is about pneumonia).

- Paragraph between lines 143 and 148 belongs to the discussion. 

- line 152 – It is worth including the operator card in the supplementary materials. 

- line 186: What do the Authors mean by "predicting the diagnosis"? Bronchiolitis diagnosis was an inclusion criterion – all the patients were diagnosed with bronchiolitis. I suppose it should be "predicting bronchiolitis severity". 

- Lines 189-193 belong to the discussion. 

Results

- Fig.2 "6 not in study time" – What do the Authors mean by that? It was not one of the exclusion criteria.

- Table 1:

-- the number of all participants is not proper (223 instead of 233)

-- Why were some characteristics not assessed in all the patients? For example, prematurity – in 230 patients. Does it mean the Authors do not know the prematurity history of 3 children? 

- line 209-210: "an early involvement of paravertebral areas was more frequent in mild cases (p 0,002)." It is not compatible with the data given in Table 2. In the last line of Table 2, the Authors reported that early involvement of the paravertebral fields was found in 62 children (84.9 %) with moderate/severe bronchiolitis and in 91 (56.9%) of children with mild bronchiolitis. If the above numbers are given properly, early involvement of paravertebral fields was more frequent in moderate/severe cases. There is no p-value given. Please, clarify this issue and complete the p-values. 

- Table 2:

-- "Lung fields number, n° (SD)" should be "Lung fields number, mean (SD)"

-- P values should be given for all the checked characteristics

- Table 3: 

-- P values should not be 0. 

-- There is no data given for the "Number of fields involved" for the "mean score model". 

- Fig. 3 - Please, use more distinctive colours. It is difficult to read the figure.

- Tab. 5 - It is very difficult to read. Please, improve it graphically so that 1 number is in 1 line, and percentage signs are together with the numbers they belong to (p.e. mild bronchiolitis n=129). If it is not possible with standard orientation, I suggest changing the page orientation. 

- Supplemental Table 1 - It is not clear what "cases with a check" mean.

- Supplementary Table 3 - Consolidation is not an artefact. In this case, you should use the term "ultrasound signs".

Discussion

- Line 290 – this is not the first multicenter study. Bueno-Campagna, cited by the Authors (ref. 22), was also multicenter. 

- "More importantly, they included a combination of artifacts, not in line with recent advances in the field of LUS/…/" Be more specific, please. For example, Bueno-Campagna et al. checked the same LUS findings as the Authors of the presented study.

- "The ultra-sound follow-up also allows us to detect any complications of a bronchiolitis which usually has a benign self-limiting course, since if after an apparent clinical / instrumental improve-ment a wors-ening of the subject's condition is observed, the ultrasound can show larger consolidations with tree-like bronchograms that can sugges ta possible bacterial complications, as suggested in other studies [19, 44]." What does it mean "instrumental improvement"? Please, rephrase for clarity. 

- "Although SVA have not been widely described in the lung ultrasound literature, their genetic basis can be predictable, essential-ly representing acoustic traps capable of small vibrational responses" What do you mean by "genetic basis of SVA"? Please, rephrase. 

- What is „bronchiolar flogosis”? This term was used neither in American guidelines nor in Italian consensus. Please, rephrase so that it is clear for the international audience. 

- Line 346: This is not the first study concerning LUS in bronchiolitis with follow-up. However, it is the first one with three control time points. 

- In the limitations, it should be mentioned that moderate and severe bronchiolitis was pulled together. 

This article would benefit from close editing. I found it difficult to follow the Authors' argument due to the many typos, improper medical nomenclature, and stylistic and grammatical errors. Most of the errors are listed below, together with suggestions for corrections:

Abstract

- Nor-mal -> normal

- There is a missing brace after the "qualitative". 

- "During follow-up, all LUS models improved (p <0,001)" – I presume that not models, but the patients, who were assessed according to the models, improved. 

Introduction

- "low-airway viral infection" –> lower respiratory tract infection

- "multicentric" –> "multi-centric" or "multicenter", as it was used by the Authors before

Materials and methods

- "the severity of bronchiolitis was cataloghyzed" – Could you rephrase it, please?

- "immunodepression" -> "children with immunosuppression" or "immunocompromised children" 

- "each participating centers" -> each participating centre

- radioloig-ic

- dis-eases

- da-ta

- commit-tees

- "The acquisitions were achieved by physicians having at least one year experience in pediatric LUS, using fundamental and harmonic images in relation to the technical characteristics of the ultrasound systems used by individual operators and in relation to the expe-rience of the operator himself." – It is not clear to the reader. Rephrase the sentence, please.

- A-Lines -> A-lines

- "linear size", "greater superficial extension" – These terms are imprecise. 

- There are unnecessary spaces in the text.

 - "increaincreasings-ing"

- "from isolated B-Lines to consolidation, through the white lung."

- ecographic -> echographic

- to evaluated -> to evaluate

Results

- Fig.2 

-- "Study drawing" is not necessary for the title. 

-- “no hospitalized” -> “not hospitalized”

- Table 2 

-- "Early involvement of paravertebral field" -> Early involvement of paravertebral fields. 

-- “intersizial” -> interstitial

- Table 3 title "Lung Ultrasound" -> lung ultrasound

- Line 263 "LUS gradually improved" -> LUS results gradually improved

- Table 5

-- title: “different follow-up” -> subsequent

Discussion

- "Since the ultrasound allows a bedside evaluation of the subpleural lung in terms of density, therefore of reduction [absolute or relative] of the quantitative ratio of the airspace and the inter-stice, it is reasonable to expect a combination of clinical and ultrasound parameters may be useful in clinical practice." – Please, rephrase, so that it is clear for English-speaking reader. 

- "worsening progression" – Rephrase, please.

- "Therefore, in our opinion the distinction between white lung and small consolidation [generally surrounded by white lung] is no such as to vary their respective scores" – It is not clear to the reader. Please, rephrase. 

- Line 342 "collapse of these areas" – This is not a proper medical term. Be more precise, please. 

- "This confirmed the general benignity of the disease and reinforces the hypothesis that much of the radiological alterations are supported by alterations of peripheral ventilation rather than to parenchymal flogosis or alveolar consolidation." – Rephrase, please.

- Line 366-368 – please, rephrase. 

References

- 2. There is a space missing between "guideline:" and "the."

- 4. in-troducing -> introducing

- 5. geno-type -> genotype, respirtory -> respiratory

 - 8. digrossi -> diagnosis 

- 10. classify-cation 

- 14. man-agement

- 15. Nuc-lear

- 21. Dolega-kozierowska

- 23. ultra-sound

- 26. Lung -> lung

SUPPLEMENTARY MATERIAL

- The used terms should be homogenous – either "supplementary" or "supplemental".

Supplemental Table 1

- „(1,2,3,level)”

Supplemental Table 2

- "breath of the nasal fins" -> "nasal flaring"

- Sato2 -> SatO2

- "episodes of the rise" – It is not proper. It should be "increasing episodes".

- Table legend contains many language errors. Correct it, please. 

Supplementary Table 3

- dicotomous -> dichotomous

- - "not normal" – Rephrase, please.

- What about consolidations of exact 1 cm size? To be precise, you should use a division: consolidation ≤ 1 cm and consolidation > 1 cm.

Supplementary Figure 1 

- Isolated B-lines – why the first one is different from the remaining ("solid")?

- A-Lines -> A-lines

- Short Vertical Artifacts -> short vertical artifacts

Author Response

REVIEWER 2

A brief summary

This observational, prospective, longitudinal, multicenter study aims to evaluate the role of lung ultrasound (LUS) in the assessment of bronchiolitis severity in children under 12 months of age. It shows the association between sonographic and clinical assessment of bronchiolitis severity short after clinical diagnosis (baseline) and at three subsequent time points during follow-up using four different LUS protocols/models (2 qualitative and 2 quantitative). It also shows the role of LUS in monitoring bronchiolitis and the diagnostic capabilities of these 4 LUS models to predict bronchiolitis severity. 

This is the first publication concerning LUS in bronchiolitis, including such a large number of patients (233 at the baseline and 185 at follow-up). To my knowledge, this is the longer follow-up time among studies published so far. LUS protocols were standardised between 12 pediatric centres. 

Thanks a lot for your appreciation and support

General concepts comments (Major criticism)

I have no major criticism.

Specific comments (Minor criticisms)

Abstract

"for the identification of severe cases of bronchiolitis". The presented study does not identify severe cases of bronchiolitis, as it assesses pulled moderate/severe cases compared to mild cases. Please rephrase it so that it represents the study's actual results. 

- "During follow-up, all LUS models improved" -> "During follow-up, LUS results according to all LUS models/protocols improved"

  • It is essential to mention in the abstract the prognostic value of LUS, which was assessed in the study. 

Thanks for the suggestion, we corrected.

Introduction

- "limited cohort of LUS signs"- What do the Authors mean by that? 

- "LUS has been proposed" - > "has been proved to be"

  • There is one more recent publication on LUS in bronchiolitis, which should be mentioned: La Regina DP, Bloise S, Pepino D, Iovine E, Laudisa M, Cristiani L, Nicolai A, Nenna R, Mancino E, Di Mattia G, Petrarca L, Matera L, Frassanito A, Midulla F. Lung ultrasound in bronchiolitis. Pediatr Pulmonol. 2021 Jan;56(1):234-239. doi: 10.1002/ppul.25156. Epub 2020 Nov 24. PMID: 33151023.

Thanks for the suggestion, we have added bibliography

Materials and methods

  • The assessment of bronchiolitis severity is not precisely described in the main text. According to Supplementary Table 2, the assessment was performed according to the modified Italian consensus method (by Baraldi et al.) and not the Baraldi method itself. Rephrase, please. I also suggest that Supplementary table 2 should be incorporated into the main text as it is crucial for understanding the methods. 

Thanks for the suggestion, we corrected. Now supplementary table 2 is table 1 of paper.

  • It is essential to describe that moderate and severe bronchiolitis are pulled together and why. 

Thanks for the suggestion, we clarified in the paper.

  • line 124-125: What about consolidations of exact 1 cm size? To be precise, a division should be used: consolidation ≤ 1 cm and consolidation > 1 cm.

Thanks for the suggestion, we corrected.

  • First Figure in the main text does not have either a title or a legend. Please, complete these elements.

Thanks for the suggestion, we corrected figure 1, we put title and legend ( now figure 2) .

- "No control cases…" – It belongs to the discussion. 

Thank you, we left it in the methods but now also added in the limitation sections of the discussion

- "Standard abdominal preset" usually contains harmonic and compound imaging, while for lung ultrasound, they should be switched off to allow assessment of artefacts. 

The reviewer is right, harmonic should be switched off during LUS assessment, however some devices may need specific interventions of bioengineering to switch off the harmonics in some US settings (as the reviewer appropriately noted), and on the same time some US settings may not be available in each centers and depend on the resources available (as well bioengineering or device techinicians from the Us brand). We are sure that the reviewer can understand that this is a difficulty when performing multicenter LUS studies, and in fact we stated in the methods “using fundamental and harmonic images in relation to the technical characteristics of the ultra-sound systems used by individual operators and in relation to the experience of the operator himself”, but on the same time we guaranteed rigorous respect of other acquisitions, like “. A 10 MHz (median frequency) linear probe, standard abdominal preset, mechanical index low-er than 0.7, intermediate gain to obtain a pleural line defined but not too saturated, unique focus on the pleural line and depth 3-5 cm were used”, as again mentioned in the text

  • What does "the first contact with the recruiting centre" mean? It is not described what a "recruiting centre" is. 

We meant “within 3-6 hours of first clinical assessment in each participating center”, we have now changed it

  • The lung fields are not clearly defined in the article's main text. Neither they are marked precisely in Supplementary Figure 1. The figure is not easy to interpret and understand. I suppose the lung fields are frontal right, frontal left, lateral right, lateral left, posterior paravertebral right, posterior paravertebral left, posterior basal right, and posterior basal left. However, it should be clear for the reader. The green and red fields representing longitudinal and transverse scans suggest that they were done on different parts of the lungs, while from the text, it is clear that transverse and longitudinal scans were both performed on each of the 8 lung fields. I suggest omitting these symbols. It is not described why paravertebral and basal areas are marked differently in the figure. I suggest omitting this check mark or explaining it. In my opinion, Supplementary Figure 1 should be incorporated into the main text, as it is crucial to understand the methods.

Thanks for the suggestion, we have corrected and put in the paper ( now figure 1). Posterior lung fields were divided into paravertebral and basal as suggested in the literature by the studies of Basile and Taveira.

- "Each step of the clinical and ultrasound evaluation involved the dichotomous attribution (presence or absence) of ultrasound signs compatible with the diagnosis of bronchiolitis, the total grading of each area explored, and the number of involved areas." – It is not clear to the reader. Rephrase the sentence, please.

We mean that for each LUS evaluation we classified images in a dichotomous way of presence/absence of artifacts suggestive of bronchiolitis. There is available literature now suggesting the most common findings in bronchiolitis, including a recent one from Bloise et al of Sapienza university of Rome that we have now included and mentioned after your kind suggestion

What does it mean that the signs were compatible with the diagnosis of bronchiolitis? All the examined patients were diagnosed with bronchiolitis. 

We mean LUS signs suggestive of bronchiolitis, according to available literature (please see our previous comment)

- Short vertical artefacts were mentioned in neither of the cited publications (30-32). I do not recognise the 3 cm depth criterion. What is its source? Short vertical artefacts are mentioned in the ref. 48 (Demi 2022).

Short vertical artifacts are mentioned more directly in reference 48 as you appropriately noted, but the physical bases of those artifacts derive from earlier observations from the same team, discussed indirectly in those mentioned papers. The 3cm depth criterion is agreed internally in our team and our referral society, as the reviewer is aware there is still lack of international standardization of LUS nomenclature and classification, but the most important thing now in a research context is that all sonographers of a team use the same classification and interpretation. Therefore, we respectfully ask the reviewer to accept our classification, although we have explained it in the limitation sections

- Intercostal space is not always equal to 1 cm, especially in small children, including infants. If you use such a criterion, you should justify it. 

Sorry we have not fully understood to what this is specifically referred, therefore could not clarify this point. Although we will be happy to do so if the reviewer still believes that we need to clarify this, as previously mentioned in LUS there are still many issues not fully standardized, and the most important part (to overcome some lack of standardization) is that all operators act in the same way

- I do not recognise the criterion "less than half cm" for multiple B-lines. What is the source of this criterion?

This is again similar to what previously mentioned, since there are no solid standardizations yet in literature regarding several LUS features, in order to be more consistent as possible, we had to clarify as much as possible what we internally meant in our team each specific LUS feature. As you mentioned, there is not solid source on that and we overcome it by rigorous internal description of each parameter. We added all these concepts in the limitation, anyway

- The paragraph between lines 129 and 136 belongs to the discussion.

Done, thanks

  • line 130: "classifications of ultrasound signs of bronchiolitis" – reference 9. does not consider bronchiolitis (it is about pneumonia).

Right, we removed reference 9 here

  • Paragraph between lines 143 and 148 belongs to the discussion. 

Changed, thank you

line 152 – It is worth including the operator card in the supplementary materials. 

  •  

As the operator card is about 12 pages long and in Italian, we kindly ask the reviewer to understand if we have not included that in the supplementary material, as it would make the file very long and not very useful to the reader

- line 186: What do the Authors mean by "predicting the diagnosis"? Bronchiolitis diagnosis was an inclusion criterion – all the patients were diagnosed with bronchiolitis. I suppose it should be "predicting bronchiolitis severity". 

Yes, thank you, now corrected

- Lines 189-193 belong to the discussion. 

Changed, thank you

Results

  • 2 "6 not in study time" – What do the Authors mean by that? It was not one of the exclusion criteria.

This 6 cases do not respect the timing set in the study design. So we excluded them.

- Table 1:

-- the number of all participants is not proper (223 instead of 233)

Thanks for the suggestion, we corrected.

-- Why were some characteristics not assessed in all the patients? For example, prematurity – in 230 patients. Does it mean the Authors do not know the prematurity history of 3 children? 

Yes, the partipating center did not provide us with some of the data in a very minority of cases

- line 209-210: "an early involvement of paravertebral areas was more frequent in mild cases (p 0,002)." It is not compatible with the data given in Table 2. In the last line of Table 2, the Authors reported that early involvement of the paravertebral fields was found in 62 children (84.9 %) with moderate/severe bronchiolitis and in 91 (56.9%) of children with mild bronchiolitis. If the above numbers are given properly, early involvement of paravertebral fields was more frequent in moderate/severe cases. There is no p-value given. Please, clarify this issue and complete the p-values. 

Sorry for the misunderstanding, the P value is included in the table section “early involvement …” and is of P< 0.001, as reported in bold in the table (which is now table 3). The lines “no” and “yes” belong both to the same data of “early involvement”

- Table 2:

-- "Lung fields number, n° (SD)" should be "Lung fields number, mean (SD)"

-- P values should be given for all the checked characteristics

Thanks for the suggestion, we corrected.

- Table 3: 

-- P values should not be 0. 

Thanks for the suggestion, we corrected.

-- There is no data given for the "Number of fields involved" for the "mean score model". 

Thanks for the suggestion, we corrected.

  • 3 - Please, use more distinctive colours. It is difficult to read the figure.

Thanks for the suggestion, we corrected and we do a new figure in according to the advice of the first reviewer.( Now figure 5)

  • 5 - It is very difficult to read. Please, improve it graphically so that 1 number is in 1 line, and percentage signs are together with the numbers they belong to (p.e. mild bronchiolitis n=129). If it is not possible with standard orientation, I suggest changing the page orientation. 

We tried to improve its readibility, thank you

  • Supplemental Table 1 - It is not clear what "cases with a check" mean.

It means cases with only the entrance check without the follow up ( in all there are 48 cases) see also figure 3  Study flow-chart.

- Supplementary Table 3 - Consolidation is not an artefact. In this case, you should use the term "ultrasound signs".

Thanks for the suggestion, we corrected.

Discussion

- Line 290 – this is not the first multicenter study. Bueno-Campagna, cited by the Authors (ref. 22), was also multicenter. 

Thanks for the suggestion, we corrected.

  • "More importantly, they included a combination of artifacts, not in line with recent advances in the field of LUS/…/" Be more specific, please. For example, Bueno-Campagna et al. checked the same LUS findings as the Authors of the presented study.

- "The ultra-sound follow-up also allows us to detect any complications of a bronchiolitis which usually has a benign self-limiting course, since if after an apparent clinical / instrumental improve-ment a wors-ening of the subject's condition is observed, the ultrasound can show larger consolidations with tree-like bronchograms that can sugges ta possible bacterial complications, as suggested in other studies [19, 44]." What does it mean "instrumental improvement"? Please, rephrase for clarity. 

- "Although SVA have not been widely described in the lung ultrasound literature, their genetic basis can be predictable, essential-ly representing acoustic traps capable of small vibrational responses" What do you mean by "genetic basis of SVA"? Please, rephrase. 

- What is „bronchiolar flogosis”? This term was used neither in American guidelines nor in Italian consensus. Please, rephrase so that it is clear for the international audience. 

- Line 346: This is not the first study concerning LUS in bronchiolitis with follow-up. However, it is the first one with three control time points. 

- In the limitations, it should be mentioned that moderate and severe bronchiolitis was pulled together. 

This article would benefit from close editing. I found it difficult to follow the Authors' argument due to the many typos, improper medical nomenclature, and stylistic and grammatical errors. Most of the errors are listed below, together with suggestions for corrections:

Thanks for the suggestion, we corrected the manuscript as you suggested below.

Abstract

- Nor-mal -> normal

- There is a missing brace after the "qualitative". 

- "During follow-up, all LUS models improved (p <0,001)" – I presume that not models, but the patients, who were assessed according to the models, improved. 

Introduction

- "low-airway viral infection" –> lower respiratory tract infection

- "multicentric" –> "multi-centric" or "multicenter", as it was used by the Authors before

Materials and methods

- "the severity of bronchiolitis was cataloghyzed" – Could you rephrase it, please?

- "immunodepression" -> "children with immunosuppression" or "immunocompromised children" 

- "each participating centers" -> each participating centre

- radioloig-ic

- dis-eases

- da-ta

- commit-tees

- "The acquisitions were achieved by physicians having at least one year experience in pediatric LUS, using fundamental and harmonic images in relation to the technical characteristics of the ultrasound systems used by individual operators and in relation to the expe-rience of the operator himself." – It is not clear to the reader. Rephrase the sentence, please.

- A-Lines -> A-lines

- "linear size", "greater superficial extension" – These terms are imprecise. 

- There are unnecessary spaces in the text.

 - "increaincreasings-ing"

- "from isolated B-Lines to consolidation, through the white lung."

- ecographic -> echographic

- to evaluated -> to evaluate

Results

- Fig.2 

-- "Study drawing" is not necessary for the title. 

-- “no hospitalized” -> “not hospitalized”

- Table 2 

-- "Early involvement of paravertebral field" -> Early involvement of paravertebral fields. 

-- “intersizial” -> interstitial

- Table 3 title "Lung Ultrasound" -> lung ultrasound

- Line 263 "LUS gradually improved" -> LUS results gradually improved

- Table 5

-- title: “different follow-up” -> subsequent

Discussion

- "Since the ultrasound allows a bedside evaluation of the subpleural lung in terms of density, therefore of reduction [absolute or relative] of the quantitative ratio of the airspace and the inter-stice, it is reasonable to expect a combination of clinical and ultrasound parameters may be useful in clinical practice." – Please, rephrase, so that it is clear for English-speaking reader. 

- "worsening progression" – Rephrase, please.

- "Therefore, in our opinion the distinction between white lung and small consolidation [generally surrounded by white lung] is no such as to vary their respective scores" – It is not clear to the reader. Please, rephrase. 

- Line 342 "collapse of these areas" – This is not a proper medical term. Be more precise, please. 

- "This confirmed the general benignity of the disease and reinforces the hypothesis that much of the radiological alterations are supported by alterations of peripheral ventilation rather than to parenchymal flogosis or alveolar consolidation." – Rephrase, please.

- Line 366-368 – please, rephrase. 

References

- 2. There is a space missing between "guideline:" and "the."

- 4. in-troducing -> introducing

- 5. geno-type -> genotype, respirtory -> respiratory

 - 8. digrossi -> diagnosis 

- 10. classify-cation 

- 14. man-agement

- 15. Nuc-lear

- 21. Dolega-kozierowska

- 23. ultra-sound

- 26. Lung -> lung

SUPPLEMENTARY MATERIAL

- The used terms should be homogenous – either "supplementary" or "supplemental".

Supplemental Table 1

- „(1,2,3,level)”

Supplemental Table 2

- "breath of the nasal fins" -> "nasal flaring"

- Sato2 -> SatO2

- "episodes of the rise" – It is not proper. It should be "increasing episodes".

- Table legend contains many language errors. Correct it, please. 

Supplementary Table 3

- dicotomous -> dichotomous

- - "not normal" – Rephrase, please.

- What about consolidations of exact 1 cm size? To be precise, you should use a division: consolidation ≤ 1 cm and consolidation > 1 cm.

Supplementary Figure 1 

- Isolated B-lines – why the first one is different from the remaining ("solid")?

- A-Lines -> A-lines

- Short Vertical Artifacts -> short vertical artifacts